# Li-Distribution in Compounds of the Li$_2$O-MgO-Al$_2$O$_3$-SiO$_2$-CaO System—A First Survey

**Thomas Schirmer** [1,*], **Hao Qiu** [2], **Haojie Li** [3], **Daniel Goldmann** [2] and **Michael Fischlschweiger** [3]

[1] Department of Mineralogy, Geochemistry, Salt Deposits, Institute of Disposal Research, Clausthal University of Technology, Adolph-Roemer-Str. 2A, 38678 Clausthal-Zellerfeld, Germany

[2] Department of Mineral and Waste Processing, Institute of Mineral and Waste Processing, Waste Disposal and Geomechanics, Clausthal University of Technology, Walther-Nernst-Str. 9, 38678 Clausthal-Zellerfeld, Germany; hao.qiu@tu-clausthal.de (H.Q.); daniel.goldmann@tu-clausthal.de (D.G.)

[3] Technical Thermodynamics and Energy Efficient Material Treatment, Institute for Energy Process Engineering and Fuel Technology, Clausthal University of Technology, Agricolastr 4, 38678 Clausthal-Zellerfeld, Germany; haojie.li@tu-clausthal.de (H.L.); michael.fischlschweiger@tu-clausthal.de (M.F.)

\* Correspondence: thomas.schirmer@tu-clausthal.de; Tel.: +49-5323-722917

**Abstract:** The recovery of critical elements in recycling processes of complex high-tech products is often limited when applying only mechanical separation methods. A possible route is the pyrometallurgical processing that allows transferring of important critical elements into an alloy melt. Chemical rather ignoble elements will report in slag or dust. Valuable ignoble elements such as lithium should be recovered out of that material stream. A novel approach to accomplish this is enrichment in engineered artificial minerals (EnAM). An application with a high potential for resource efficient solutions is the pyrometallurgical processing of Li ion batteries. Starting from comparatively simple slag compositions such as the Li-Al-Si-Ca-O system, the next level of complexity is reached when adding Mg, derived from slag builders or other sources. Every additional component will change the distribution of Li between the compounds generated in the slag. Investigations with powder X-Ray diffraction (PXRD) and electron probe microanalysis (EPMA) of solidified melt of the five-compound system Li$_2$O-MgO-Al$_2$O$_3$-SiO$_2$-CaO reveal that Li can occur in various compounds from beginning to the end of the crystallization. Among these compounds are Li$_{1-x}$(Al$_{1-x}$Si$_x$)O$_2$, Li$_{1-x}$Mg$_y$(Al)(Al$_{3/2y+x}$Si$_{2-x-3/2y}$)O$_6$, solid solutions of Mg$_{1-(3/2y)}$Al$_{2+y}$O$_4$/LiAl$_5$O$_8$ and Ca-alumosilicate (melilite). There are indications of segregation processes of Al-rich and Si(Ca)-rich melts. The experimental results were compared with solidification curves via thermodynamic calculations of the systems MgO-Al$_2$O$_3$ and Li$_2$O-SiO$_2$-Al$_2$O$_3$.

**Keywords:** lithium; thermodynamic modeling; engineered artificial minerals (EnAM); melt experiments; PXRD; EPMA

## 1. Introduction

With respect to the development in electromobility as well as to the changes in circular energy systems, Li-ion batteries are of central importance. To safeguard raw material sources especially for critical elements such as Co, Ni and Li as key components of this technology, efficient recycling processes are essential. One of the most important routes to recycle these battery types is the pyrometallurgical processing, which can deal with a broad range of input material. While Co, Ni and other valuable heavy metals such as Cu report into the alloy melt, Li is transferred at least in major amounts into the slag phase of this process.

A resource and energy efficient recovery of Li from the slag could be accomplished, if Li were concentrated in specific Li-rich artificial mineral phases, which could then be separated after crystallization and cooling of the slag by means of mineral processing technologies, generating concentrates for following hydrometallurgical processing.

Previous research has shown that Li can be recovered in the form of the $LiAlO_2$ crystals through flotation from a remaining silicate slag matrix [1]. The hydrometallurgical processing of Li enriched silicate slag has also shown that Li recovery can reach 80–95% [2].

As long as the complete system, based on a $Li_2O$-$Al_2O_3$-$SiO_2$-$CaO$ mixture, does not contain any other element, the results are promising. As soon as other elements are added, new phases start to crystallize.

Besides Li and Al, reporting from the Li-ion battery input into the slag, Si, Ca and often Mg (at least partly from dolomite as slag building component) are introduced as slag builders to ensure an optimized split between metal alloy melt, slag and dust phase in the pyrometallurgical process.

Until now, the thermodynamics of the overall process have not been investigated sufficiently and therefore for extended systems such as $Li_2O$-$MgO$-$Al_2O_3$-$SiO_2$-$CaO$ this work serves as a starting point. Consequently, this should allow understanding some basic principles and giving further insights into these slag-systems. Additionally, a solid ground should be provided for further research on these slag systems, because in the future more complex slag systems, e.g., Mn-containing mixtures, should be investigated since they will represent future inputs to this recycling route especially for the NCM-type batteries.

The Umicore Battery Recycling Process is a vital pyrometallurgical process developed for the recovery of NiMH and spent lithium-ion batteries [3]. From the composition of a slag with the compounds $Li_2O$-$MgO$-$Al_2O_3$-$SiO_2$-$CaO$ and high aluminum content, it is observed that Li is present in the slag in the form of the $LiAlO_2$ [2], which would facilitate subsequent recovery by flotation. At the same time, the spinel phase appears in all three Umicore slags, and, in one of the Umicore slags, Li is even partially dispersed in the spinel phase [3].

Even though spinel phases appear in different slags if bivalent ions such as those of Mg are present, there is little published research on the impact of spinel on the formation of separate $LiAlO_2$ crystals because of the scavenging of Al from the melt and the formation of $Mg_{1-(3/2y)}Al_{2+y}O_4$/$LiAl_5O_8$ solid solution.

In this study, three synthetic slags with different contents of MgO based on the $Li_2O$-$MgO$-$Al_2O_3$-$SiO_2$-$CaO$ oxide system were prepared using pure chemical reagents. The degree of supercooling was then reduced by controlling and cooling the melt slowly to obtain thoroughly crystallized synthetic slags for research. The synthetic slags were then analyzed by X'Ray powder diffraction (PXRD) and electron probe microanalysis (EPMA) for mineralogical studies and finally compared to the solidification curves obtained by thermodynamic calculation. This served as a starting point for studying the influence of spinel formation and understanding important phase reactions in the five-component oxide system Li–Mg–Al–Si–Ca.

To increase the knowledge on the behavior of slag systems and the options to predict and stimulate the creation of artificial mineral phases, an interdisciplinary approach was taken, comprising thermodynamical modeling, pyrometallurgical processing, mineralogical analysis and prediction and testing of mineral processing technologies. In this paper, the focus is put on mineralogical analysis in connection with thermodynamic modeling.

## 2. Background

To better understand the results presented in this article, the existing information about the compounds of important binary and ternary systems containing $Li_2O$, $MgO$, $Al_2O_3$, $SiO_2$ and $CaO$ is summarized. This information serves as the starting point to analyze and improve the existing data and develop respective thermodynamic modeling strategies.

*2.1. Important Binary Phase Systems Containing Li*

In the systems $Li_2O$-$CaO$ and $Li_2O$-$MgO$, except for limited solid solution, no explicit phase reactions are reported (e.g., Konar et al. [4]).

In the system $Li_2O$-$Al_2O_3$, several stable lithium aluminate compounds are described: $Li_5AlO_4$, $LiAlO_2$ and $LiAl_5O_8$ [5,6]. Additionally, a high temperature compound $LiAl_{11}O_{17}$ at $0.8 < Al_2O_3 < 0.92$ and $>2200 \,°C$ is mentioned [5]. The compounds $Li_2Al_4O_7$ synthesized by Kale et al. [7] and $Li_3AlO_3$ were found to be instable by Kale et al. [7] and are not part of the data published by Konar et al. [5]. In this phase diagram, there is also a thermal barrier at the mole fraction of $Al_2O_3 = 0.5$ ($LiAlO_2$), so that at $0.18 < Al_2O_3 < 0.5$ the resulting mixture is $Li_5AlO_4$/$LiAlO_2$ and at $0.5 < Al_2O_3 < 0.82$ the resulting mixture is $LiAlO_2$/$LiAl_5O_8$. The two compounds important for this work, $LiAlO_2$ and $LiAl_5O_8$, both have polymorphs. According to Konar et al. [5], $LiAlO_2$ comprises a tetragonal $\gamma$-phase (high temperature) and a cubic $\alpha$-phase (low temperature) modification and $LiAl_5O_8$ generally crystallizes in a spinel (high temperature) and a low temperature primitive cubic form [8]. According to Li et al. [9], $LiAlO_2$ comprises four polymorphs: a tetragonal $\gamma$-phase, a rombohedral $\alpha$-phase, an orthorhombic $\beta$-phase and two phases of high temperature.

In the system $MgO$-$Al_2O_3$, the only binary compound is cubic $MgAl_2O_4$ (spinel) with the idealized composition at a mole fraction of $Al_2O_3 = 0.5$. At this ratio, there is also a thermal barrier. In the area of mole fraction $0 < MgO < 0.05$ in the temperature range 1900–2800 $°C$, solid $MgO$ can form a solid solution with $Al_2O_3$ [5]. The region of mole fraction $0.5 < Al_2O_3 < 0.96$, particular important for this study, comprises a complete solid solution, so that an Al-rich melt can be in equilibrium with a spinel relatively enriched in Mg [10].

The system $Li_2O$-$SiO_2$ comprises the binary compounds $Li_8SiO_6$, $Li_4SiO_4$, $Li_6Si_2O_7$, $Li_2SiO_3$ and $Li_2Si_2O_5$ [11]. Additionally, the prediction shows two thermal barriers at the composition $Li_4SiO_4$ and $Li_2SiO_3$.

*2.2. Important Ternary Phase Systems Containing Li*

In the system $Li_2O$-$MgO$-$Al_2O_3$, three important primary crystallization fields can be predicted [5]: spinel ($MgAl_2O_4$), $MgO$ and $\gamma$-$LiAlO_2$. Interesting isopleths are spinel-$LiAl_5O_8$, spinel-$LiAlO_2$, spinel-$Li_2O$ and $MgO$-$LiAlO_2$. From this intersects, it can be concluded that a limited amount (i.e., maximum mole fraction = 0.31) of $LiAlO_2$ can be dissolved in $MgO$. Additionally, the compounds $LiAl_5O_8$ and $MgAl_2O_4$ can be combined to an ideal spinel solid solution [5].

The system $Li_2O$-$Al_2O_3$-$SiO_2$ contains the Li-bearing binary systems $Li_2O$-$SiO_2$ and $Li_2O$-$Al_2O_3$, as described in Section 2.1 [12]. With respect to the present work, the primary crystallization fields of $LiAlO_2$, $LiAl_5O_8$, eucryptite ($LiAlSiO_4$) and spinel are of particular interest. The compound $LiAlO_2$, described in Section 2.1, can additionally incorporate Si according to an substitution of $Li^+ + Al^{3+} = Si^{4+} + v$ (vacancy) so that the general formula is $\alpha$ ($LiAl^{4+}$, $vSi^{4+}$])$O_2$ and $\gamma$ ($Li$, $v$)$^{Li}$[$Al^{3+}$, $Si^{4+}$]$^M O_2$ [12]. The compound eucryptite can be derived from $SiO_2$ via a substitution of $Li^+ + Al^{3+} = Si^{4+} + v$ [12] and crystallizes as quartz in the trigonal system, whereas a low temperature $\alpha$-polymorph is disordered and a $\beta$-polymorph is ordered. Additionally, this compound can incorporate Mg and be broken up into the compounds $LiAlO_2$, $Mg_{0.5}AlO_2$ and $SiO_2$ [13]. The spinel crystallizes in a cubic system and can have a very variable chemistry with respect to the Al/Mg ratio and the solid solution with $LiAl_5O_8$ (see Section 2.1).

# 3. Materials and Methods

*3.1. Materials*

Chemicals

The chemicals used for producing synthetic slags are lithium carbonate (Merck, purum), calcium oxide (Sigma-Aldrich, reagent grade, St. Louis, MO, United States), silicon dioxide (Sigma-Aldrich, purum p.a.,

St. Louis, MO, United States), aluminum oxide (Merck) and magnesium oxide (98% wt.%, Roth, Karlsruhe, Germany). All chemicals ordered via Merck KGaA, Darmstadt, Germany.

### 3.2. Methods

#### 3.2.1. Experiments

The chemical compositions of input materials for the synthesis of slags are listed in Table 1. The chemicals were manually mixed in a mortar and grinded in a disc mill for 5 min. Each sample was placed in a Pt-Rh crucible and heated in a chamber furnace (Nabertherm HT16/17, Nabertherm GmbH, Lilienthal, Germany) in an air atmosphere. The heating regime is shown in Figure 1. A heating rate of 2.89 °C/min was first employed to reach 720 °C, which is the melting temperature of $Li_2CO_3$, and then a heating rate of 1.54 °C/min was used to aid in the decomposition of $Li_2CO_3$ and to reach the target temperature. Samples were kept at 1600 °C for 2 h. Thereafter, the samples were cooled to 500 °C at a cooling rate of 0.38 °C/min and quenched in water.

**Table 1.** Calculated Theoretical Chemical Bulk Composition of the Samples According to the Weighed Quantities.

| Sample | $Li_2CO_3$ | $CaCO_3$ | $SiO_2$ | $Al_2O_3$ | $MgO$ |
|---|---|---|---|---|---|
| Content | % | % | % | % | % |
| 1 | 22.87 | 22.40 | 16.36 | 32.97 | 5.40 |
| 2 | 22.51 | 21.55 | 16.08 | 32.17 | 7.70 |
| 3 | 22.32 | 21.24 | 15.63 | 31.54 | 9.27 |

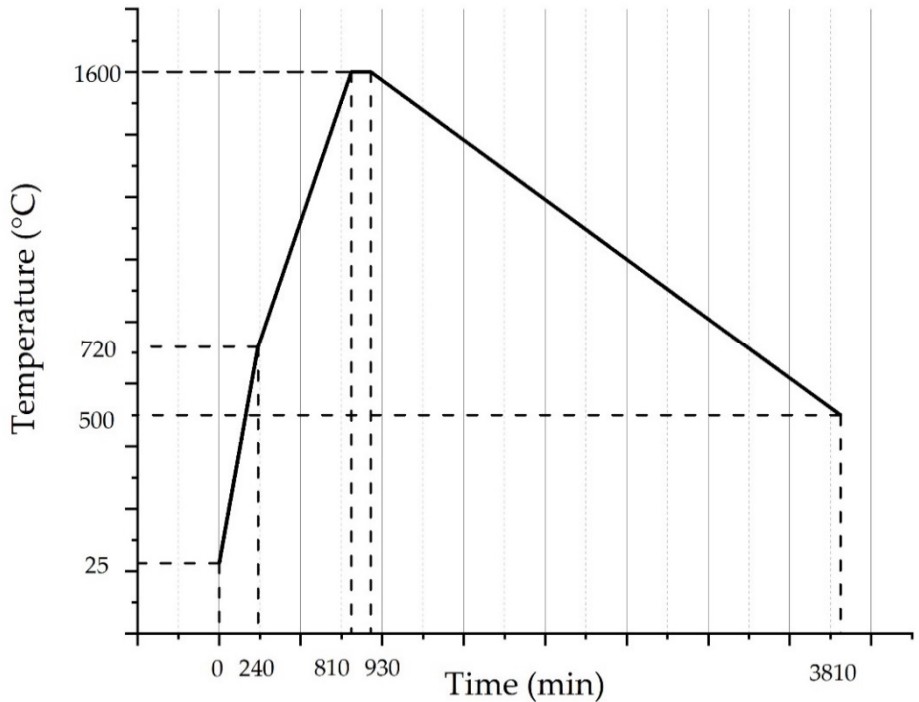

**Figure 1.** Schematic diagram of temperature regime.

#### 3.2.2. Chemical Bulk Analysis

The element content was determined with ICP optical emission spectrometry (ICP-OES 5100, Agilent, Agilent Technologies Germany GmbH & Co. KG, Waldbronn, Germany). Samples were melted with lithium tetra borate in a platinum crucible at 1050 °C, and then the samples were leached with dilute hydrochloric acid to measure the content of Al, Ca, Mg and Si. To measure other elements,

the samples were mixed with nitric acid and digested at 250 °C and under a pressure of 80 bar in an autoclave (TurboWAVE, MLS, Leutkirch im Allgäu, Germany).

### 3.2.3. Mineralogical Investigation

An overview of the mineralogical composition was provided by powder X-Ray diffraction (PXRD), using a PANalytical X-Pert Pro diffractometer, equipped with a Co-X-Ray tube (Malvern Panalytical GmbH, Kassel, Germany). For identification of the compounds, the pdf-2 ICCD XRD database, the American Mineralogist Crystal Structure Database [14] and the RRUFF-Structure database [15] were assessed.

The analysis of single crystals and grains was carried out with electron probe microanalysis (EPMA). EPMA is a standard method to characterize the chemical composition in terms of single spot analysis or element distribution patterns, accompanied by electron backscattered Z (ordinal number) contrast (BSE(Z)) or secondary electron (SE) micrographs. To carry out EPMA measurements, the sample was prepared as polished block in epoxy resin, coated with carbon and characterized using a Cameca SX$^{FIVE}$ FE Field Emission) electron probe, equipped with five wavelength dispersive (WDX) spectrometers (CAMECA SAS, Gennevilliers Cedex, France). The following elements/(lines) were used to quantify the measurement points: Na (K$\alpha$), Mg (K$\alpha$), Al (K$\alpha$), Si (K$\alpha$), K (K$\alpha$), Ca (K$\alpha$), Ti (K$\alpha$), Mn (K$\alpha$) and Fe (K$\alpha$). To calibrate the wavelength dispersive X-ray fluorescence spectrometers (WDRFA), an appropriate suite of standards and analyzing crystals was used. The reference materials were provided by P&H Developments Ltd. (Glossop, Derbyshire, UK) and Astimex Standards Ltd. (Toronto, ON, Canada). The beam size was set to 0, leading to a beam diameter of substantially below 1 μm (100–600 nm with field emitters of Schottky-type, e.g., Jercinovic et al. [16]). To evaluate the measured intensities, the X-PHI-Model was applied [17].

Lithium, one of the key elements in this study, cannot be directly analyzed since EPMA uses X-ray fluorescence to detect the elements in the sample and the extremely low fluorescence yield and long wavelength of Li K$\alpha$ makes the direct determination of this element nearly impossible. With the reasonable assumption that other refractory light elements such as Be and B are not present in the investigated material and volatile elements and compounds such as F, $H_2O$, $CO_2$ or $NO_3^-$ are effectively eliminated during the melt experiment, Li can be calculated using virtual compounds, as depicted in as described in Section 4.3.1. If necessary, the balanced Li concentration was included into the matrix correction calculation. To access the analytical accuracy with respect to Li-containing compounds, the international reference material spodumene (Astimex) and the in-house standard $LiAlO_2$ were used (Table 2). Additionally, Li containing crystalline phases identified by X-ray diffraction (PXRD) could be referenced to the EPMA result.

**Table 2.** Repeated Measurements on Two Li-Compounds. Spod, Spodumene; %StdDev, Percentage Standard Deviation of the Measured Points (pepeats: *n* = 5); R, Recovery; LiAl, $LiAlO_2$.

| wt.% | Average Spod. | %StdDev, Spod. | Ref. Spod. | R (%) | Average LiAl | %StdDev, LiAl | Ref. LiAl | R (%) |
|---|---|---|---|---|---|---|---|---|
| **Al** | 15.04 | 0.35 | 14.4 | 104 | 41.24 | 0.22 | 40.9 | 101 |
| **Mg** | 0.00 | n. a. | 0.0 | n. a. | 0.01 | n. a. | 0.0 | n. a. |
| **Ti** | 0.00 | n. a. | 0.0 | n. a. | 0.00 | n. a. | 0.0 | n. a. |
| **Mn** | 0.05 | n. a. | 0.0 | n. a. | 0.00 | n. a. | 0.0 | n. a. |
| **Fe** | 0.02 | n. a. | 0.0 | n. a. | 0.03 | n. a. | 0.0 | n. a. |
| **Ca** | 0.01 | n. a. | 0.0 | n. a. | 0.01 | n. a. | 0.0 | n. a. |
| **K** | 0.00 | n. a. | 0.0 | n. a. | 0.00 | n. a. | 0.0 | n. a. |
| **Si** | 28.71 | 0.56 | 30.0 | 96 | 0.01 | n. a. | 0.0 | n. a. |
| **Na** | 0.10 | 2.83 | 0.09 | 112 | 0.00 | n. a. | 0.0 | n. a. |

### 3.2.4. Thermodynamic Modeling

For a better understanding of the experimental mechanisms investigated in the $Li_2O$-MgO-$Al_2O_3$-$SiO_2$-CaO system, the thermodynamic modeling of the phase behavior and the solidification in subsystems is of high relevance and hence applied in this work. Especially the knowledge of the phase behavior of the MgO-$Al_2O_3$ subsystem and the phases solidified at respective temperatures of certain component concentrations of the $Li_2O$-$Al_2O_3$-$SiO_2$ subsystem is important and contributes to the clarification and understanding of primary crystallization mechanisms figured out by the mineralogical characterization. On principal, based on already existing experimental data and thermodynamic studies, which are stated below, an optimized database for the subsystem was completed and applied to calculate the respective phase and solidification behavior. Generally, all calculations, i.e., for the binary MgO-$Al_2O_3$ and the ternary $Li_2O$-$Al_2O_3$-$SiO_2$ subsystems, were performed with the modified quasi-chemical model (MQM) [18–20] and the compound-energy formalism (CEF) [21], implemented in Factsage [22].

Specific insights into the database adaption regarding the two subsystems are presented subsequently. The thermodynamic database for the oxides such as MgO and $Al_2O_3$ comes from the FT oxide database [22] without any modification. Regarding the ternary subsystem $Li_2O$-$Al_2O_3$-$SiO_2$, the thermodynamic properties of $SiO_2$, $Al_2O_3$ and the mullite solid solution were used from the FT oxide [22] database without any modification. However, for the Gibbs energy of the $Li_2O$, the optimized value from [11] was integrated into the database. For compounds such as $Li_2SiO_3$, $Li_4SiO_4$, $Li_6Si_2O_7$, $Li_2Si_2O_5$-LT (low-temperature form) and $Li_2Si_2O_5$-HT (high-temperature form), the thermodynamic data were taken from [11]. The standard formation enthalpy of $Li_8SiO_6$ was optimized in this work with a value of 3, 521, 499.2 J/mol. Furthermore, for the binary compounds in the $Li_2O$-$Al_2O_3$, the standard formation enthalpy of the $Li_5AlO_4$ was optimized to 2,389,980 J/mol. The standard entropy of $LiAl_{11}O_{17}$ was optimized to a value of 350.55 $Jmol^{-1}K^{-1}$. The ternary compounds including the $\alpha$- and $\beta$-eucryptite solid solutions, $\beta$-spodumene solid solution and $\alpha$-$LiAlO_2$ solid solution were obtained from [12] without any modification. However, the Gibbs energy of the end member $G^0_{VaAlO_2}$ in the $\gamma$-$LiAlO_2$ solid solution was calculated with the assumption that the reciprocal energy of endmember is zero, while the other three endmembers were obtained directly from [12].

Based on these data, the CALPHAD calculations were performed for the subsystems, which are used for further explanations and discussions in connection with the new experimental findings in the next section.

## 4. Results

This section presents the measurement results of the melt experiments from PXRD and EPMA. First, three PXRD measurements from experiments with different Mg-concentration are compared (Section 4.2). In Section 4.3, an overview of the material with BSE(Z) micrographs and detailed spatially resolved quantitative point measurements and element distribution profiles recorded with EPMA are presented. Additionally, in Section 4.4, experimental findings are compared with thermodynamic model predictions for the relevant subsystems.

### *4.1. Bulk Chemistry of the Melt Experiments*

The measurement results presented in Table 3 show that 14–20% of Li is lost during the melting and cooling of the material. The same applies for Na, which always appears as contaminant in open systems due to the overall availability (air, dust, skin, clothing, etc.).

**Table 3.** Comparison of the Bulk Chemical Composition Measured with ICP-OES of the Four Melt Experiments, Given in Mole Percent. The bold emphazises the Li-loss which is important to see (Li is volatile).

|  | Raw Mix (Mole Fraction) | | | Product (Mole Fraction) | | | Recovery % | | |
|---|---|---|---|---|---|---|---|---|---|
|  | **V1** | **V2** | **V3** | **V1** | **V2** | **V3** | **V1** | **V2** | **V3** |
| **$Al_2O_3$** | 32.90 | 32.10 | 31.48 | 34.21 | 33.64 | 33.63 | 3.99 | 4.79 | 6.86 |
| **CaO** | 22.35 | 21.50 | 21.19 | 22.79 | 22.57 | 22.13 | 1.97 | 4.95 | 4.44 |
| **$Li_2O$** | 22.81 | 22.46 | 22.28 | 20.00 | 18.69 | 17.57 | **−12.35** | **−16.77** | **−21.11** |
| **MgO** | 5.32 | 7.59 | 9.14 | 5.32 | 8.27 | 9.90 | −0.11 | 9.03 | 8.32 |
| **$SiO_2$** | 16.32 | 16.05 | 15.60 | 17.43 | 16.59 | 16.53 | 6.81 | 3.40 | 5.98 |
| **$Na_2O$** | 0.3 | 0.3 | 0.3 | 0.3 | 0.23 | 0.2 | −14.0 | −21.4 | −28.6 |

## 4.2. PXRD Comparison of the Three Melt Experiments

The results of the PXRD measurements are presented in Figure 2, showing an overview of the diffractograms of all experiments (above) and three enlarged sections, showing important spinel and lithium aluminate diffraction peaks.

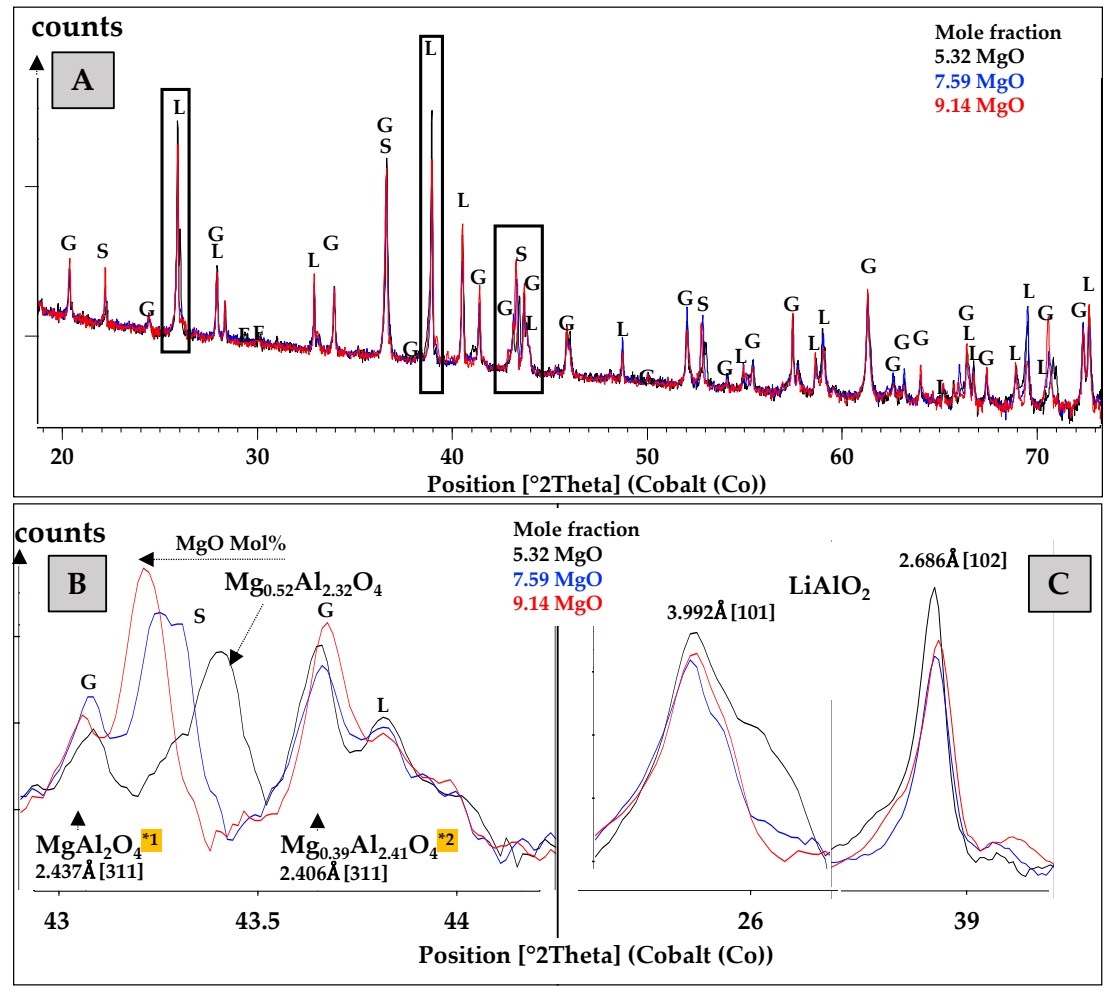

**Figure 2.** (**A**) PXRD of the solidified melt. G, gehlenite; S, spinel; L, $LiAlO_2$; E, eucryptite. (**B**) Enlarged section of the main spinel peak, * 1,: position of the main peak of $MgAl_2O_4$ from the ICCD-PDF2 No. 00-021-1152; * 2, position of the main peak of an Al-rich spinel from the ICCD-PDF2 No. 00-048-0528, peaks; $Mg_{0.52}Al_{2.32}O_4$, average composition of a spinel grain of the melt experiment with 5.32 Mol% Mg, determined with EPMA (see Section 4.2). (**C**) Enlarged sections of the first two main $LiAlO_2$ peaks.

The overview of all XRD measurements show the compounds gehlenite, spinel, $LiAlO_2$ and eucryptite (Figure 2A), whereas eucryptite is at the detection limit (<2–5 wt.%). The enlarged section of the 2-theta region of the main spinel peaks gives an indication of the changing composition of the spinel with the change of the Mg content (Figure 2B). Because of the high Al-concentration, the main (100%) spinel peaks of all experiments lie between those of the standard spinel $MgAl_2O_4$ and an aluminum-dominated $Mg_{1-(3/2y)}Al_{2+y}O_4$. Additionally, there is an indication of increasing spinel content with rising Mg concentrations. The Li-Al-oxide peaks are best explained with the diffraction pattern of $LiAlO_2$ (ICCD PDF2 No. 00-038-1464). The comparison of the two main peaks of the three experiments gives a hint that the amount of crystalline $LiAlO_2$ could be negatively correlated with the amount of Mg in the melt because the highest main peak intensities were measured in the sample with the lowest Mg concentration (Figure 2C).

*4.3. EPMA Results*

The main compounds of the melt experiments, determined with EPMA, were:

- Spinel: $Mg_{1-(3/2y)}Al_{2+y}O_4$
- Lithium aluminate (LiAl): $Li_{1-x}(Al_{1-x}Si_x)O_2$
- Eucryptite-like lithium alumosilicate (ELAS): $Li_{1-x}Mg_y(Al)(Al_{3/2y+x}Si_{2-x-3/2y})O_6$
- Gehlenite-like calcium-alumosilicate (GCAS): $Ca_2AL_2SiO_7$ with minute amounts of Mg

The compound (GCAS) is an end member of the melilite-like calcium-alumosilicate (MCAS), which is used for this phase with higher amounts of ions in addition to Ca:

- Melilite-like calcium-alumosilicate (MCAS): $(Na,Ca,Li)_2(Al,Mg,Li)(Al,Si)_2O_7$, which according to the calculations (Section 4.3.3) can also be a potential host for Li

An overview recorded with BSE(Z) shows a matrix of bright Ca-alumosilicate (GCAS/MCAS) interspersed with dendritic or massive dark LiAl. Within this mixture, large idiomorphic or hypidiomorphic crystals of spinel can be identified (Figures 3 and 4).

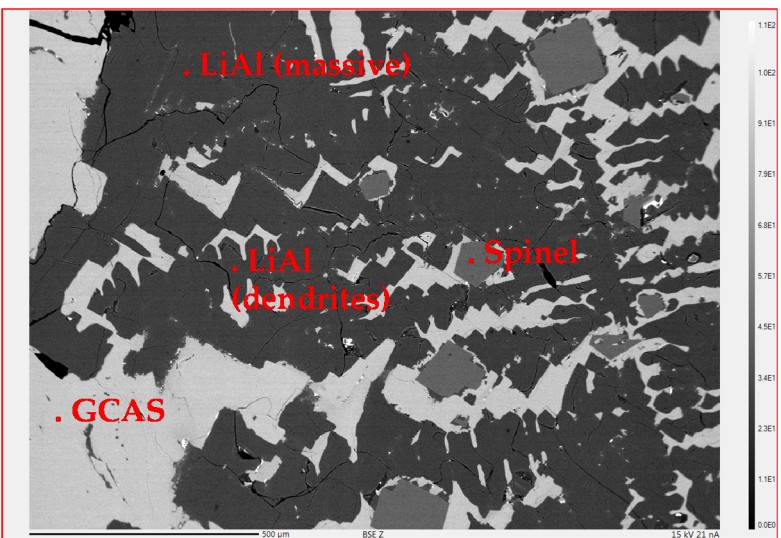

**Figure 3.** Electron micrograph (BSE(Z) of the solidified melt. Medium grey grains, spinel; dark gray sections and dendrites, LiAl surrounded by Ca-alumosilicate (GCAS, light grey sections); black, pores or preparation damage.

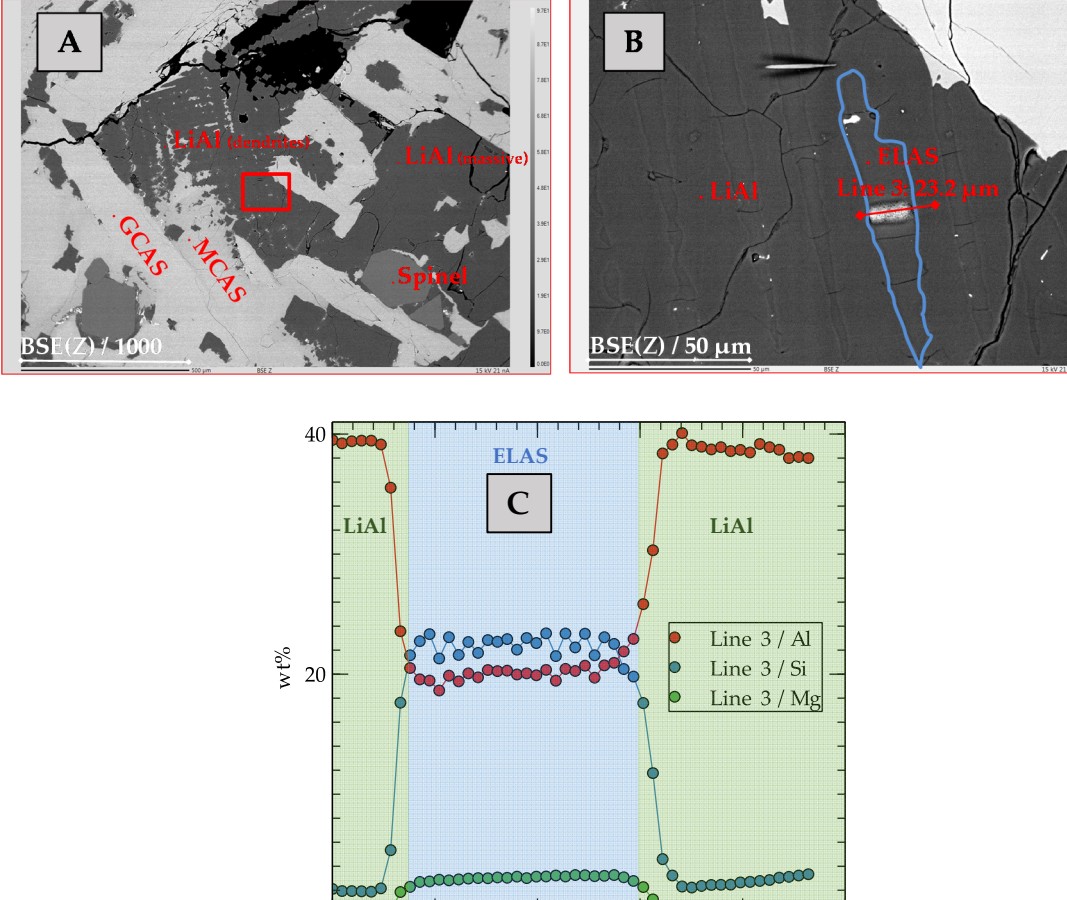

**Figure 4.** (**A**): Overview of the solidified melt (backscattered electron micrographs BSE(Z)). Medium grey grains, spinel; dark gray sections and dendrites, LiAl surrounded by Ca-alumosilicate (GCAS, MCAS, light grey sections); black, pores or preparation damage; red square, detail presented in (**B**). (**B**) Enlarged section from the red square in (**A**): the blue rim marks the grain of ELAS where the scan of Line 3 (red line) was measured. (**C**) Quantitative line scan of Line 3 (red line in (**B**)).

### 4.3.1. Lithium Aluminate (LiAl) and Lithium-Alumosilicate (ELAS)

The LiAl can be classified into two morphologic forms: massive and dendrite-like (Figures 3 and 4A). A closer look into the massive LiAl reveals thin lath-shaped grains of ELAS or a corresponding melt (Figure 4B). A line scan over such a lath-shaped grain reveals a quite homogeneous composition with more or less sharp borders to the surrounding LiAl (Figure 4C). The ELAS can be described as a mixture of the virtual compounds $LiAlO_2$, $Mg_{0.5}AlO_2$ and $SiO_2$, as listed in Table 4.

Mult. depicts a factor to multiply the three components to generate an optimized ELAS or $Li_{1-x}Mg_y(Al)(Al_{3/2y+x}Si_{2-x-3/2y})O_6$ similar to the average measured concentrations (except for Li) on Line 3 (Figure 4). The Li value results from the multiplications and this was used to calculate the formula of the analyzed ELAS in the sample. In a similar manner, the Si-containing LiAl with the general formula $Li_{1-x}(Al_{1-x}Si_x)O_2$ can be calculated as a mixture of $SiO_2$ and $LiAlO_2$. The calculated formulas of the ELAS and the LiAl are:

$$ELAS: (Li_{0.96}Mg_{0.24})(Al)(Al_{0.45}Si_{1.55})O_6$$

$$LiAl: (Li_{0.94})(Al_{0.94}Si_{0.06})O_2$$

The Si concentration in the dendritic LiAl is distinctively lower as in the massive crystals (compare Tables 4 and 5). The calculated formula of the LiAl in this case is:

$$\text{LiAl: } (Li_{0.97})(Al_{0.97}Si_{0.03})O_2$$

**Table 4.** Calculation of virtual compound ratios and average composition of the ELAS and the LiAl on Line 3, shown in Figure 4. Opt., calculated ideal composition; Meas., measured average; Mult., factor for multiplication of the virtual compounds; (Calc.), calculated values (Li, O); %StdDev, percentage standard deviation of the measured points (LiAl (Meas.), repeats, $n = 23$).

| wt.% | Virtual Compounds | | | ELAS (Opt.) | ELAS, Meas. | ELAS %StDev. | LiAl (Opt.) | LiAl (Meas.) | LiAl %StDev. |
|---|---|---|---|---|---|---|---|---|---|
| | $LiAlO_2$ | $Mg_{0.5}AlO_2$ | $SiO_2$ | | | | | | |
| Al | 40.9 | 37.9 | 0.0 | 20.1 | 20.1 | 3.2 | 38.9 | 38.9 | 1.4 |
| Mg | 0.0 | 17.1 | 0.0 | 3.1 | 3.0 | 7.5 | 0.0 | 0.0 | n. a. |
| Ti | 0.0 | 0.0 | 0.0 | 0.0 | 0.0 | n. a. | 0.0 | 0.0 | n. a. |
| Mn | 0.0 | 0.0 | 0.0 | 0.0 | 0.1 | n. a. | 0.0 | 0.0 | n. a. |
| Fe | 0.0 | 0.0 | 0.0 | 0.0 | 0.1 | n. a. | 0.0 | 0.1 | n. a. |
| Ca | 0.0 | 0.0 | 0.0 | 0.0 | 0.0 | 8.9 | 0.0 | 0.0 | n. a. |
| K | 0.0 | 0.0 | 0.0 | 0.0 | 0.0 | n. a. | 0.0 | 0.0 | n. a. |
| Si | 0.0 | 0.0 | 46.7 | 23.2 | 22.4 | 3.6 | 2.5 | 2.5 | 18.3 |
| Na | 0.0 | 0.0 | 0.0 | 0.0 | 0.0 | n. a. | 0.0 | 0.0 | n. a. |
| O (Calc.) | 48.5 | 45.0 | 53.3 | 50.2 | 49.5 | n. a. | 49.0 | 49.1 | n. a. |
| Li (Calc.) | 10.5 | 0.0 | 0.0 | 3.4 | 3.4 | n. a. | 10.01 | 10.01 | n. a. |
| Mult. | 0.33 | 0.18 | 0.49 | ← Multiplication factors for ELAS (Opt.) | | | | | |
| Mult. | 0.95 | 0 | 0.049 | ← Multiplication factors for LiAl (Opt.) | | | | | |
| Sum | 100 | 100 | 100 | 100 | 98.7 | | 100.4 | 100.6 | |

**Table 5.** Calculation of Virtual Compound Ratios and Average Composition of the LiAl in the Dendrites (Dend.) Shown in the BSE(Z) Micrograph of Figure 3. Opt., Calculated Ideal Composition; Meas., Measured Average; Mult., Factor for Multiplication of the Virtual Compounds; (Calc.), Calculated Values (Li, O); %StdDev, Percentage Standard Deviation of the Measured Points (LiAl (Dend.) (Meas.), Repeats, $n = 4$).

| wt.% | Virtual Compounds | | LiAl (Opt.) | LiAl (Dend.) (Meas.) | LiAl (Dend.) % StDev. |
|---|---|---|---|---|---|
| | $LiAlO_2$ | $SiO_2$ | | | |
| Al | 40.9 | 0.0 | 40.3 | 40.3 | 0.4 |
| Mg | 0.0 | 0.0 | 0.0 | 0.0 | n. a. |
| Ti | 0.0 | 0.0 | 0.0 | 0.0 | n. a. |
| Mn | 0.0 | 0.0 | 0.0 | 0.0 | n. a. |
| Fe | 0.0 | 0.0 | 0.0 | 0.0 | n. a. |
| Ca | 0.0 | 0.0 | 0.0 | 0.0 | n. a. |
| K | 0.0 | 0.0 | 0.0 | 0.0 | n. a. |
| Si | 0.0 | 46.7 | 1.1 | 1.1 | 12.3 |
| Na | 0.0 | 0.0 | 0.0 | 0.0 | n. a. |
| O (Calc.) | 48.5 | 53.3 | 49.0 | 49.1 | n. a. |
| Li (Calc.) | 10.5 | 0.0 | 10.36 | 10.36 | n. a. |
| Mult. | 0.98 | 0.024 | ← Multiplication factors for LiAl (Opt.) | | |
| Sum | 100 | 100 | 100.8 | 100.9 | |

### 4.3.2. Spinel

Spinel as the first crystallizing compound obeys the crystallization equilibrium inasmuch as the composition of the spinel with the highest Al content is connected with the corresponding Al-rich melt. The Mg concentrations in the measured profile (Figure 5B) are increasing from the center to the rim of

the crystal. A look at the ratio of Mg/Al in a line scan through a spinel crystal starting at the center of the grain in the direction to the rim shows no increase within a first region. After this first region, the ratio increases. Closer to the rim, the ratio decreases sharply and directly at the rim (a few µm) the ratio development of the two elements is reversed again (Figure 5B).

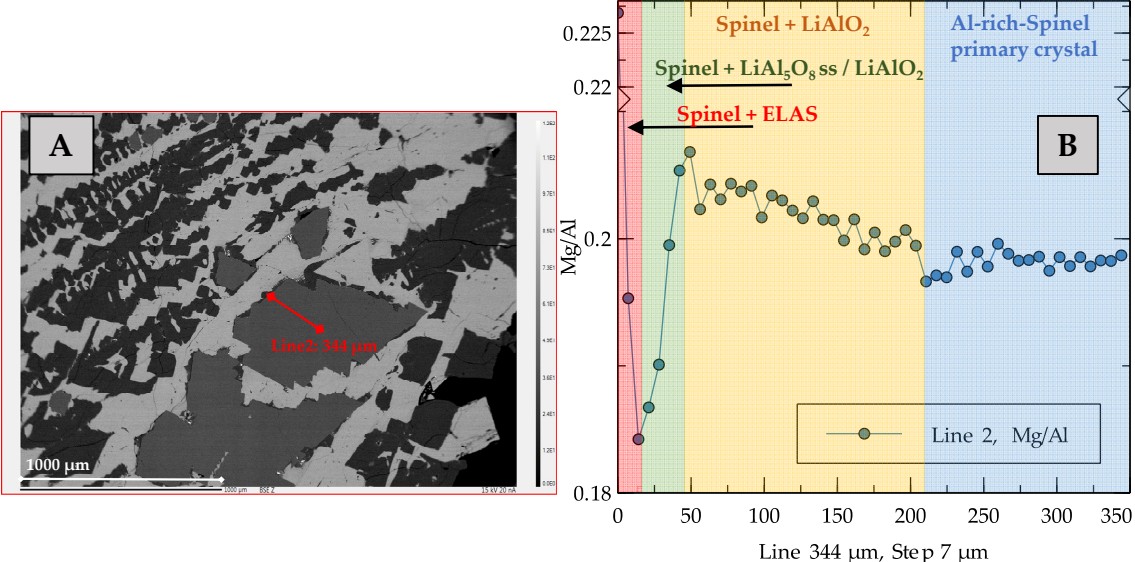

**Figure 5.** (**A**): Electron micrograph of a spinel crystal (medium grey), partly with a thin coat of LiAl (dark grey sections) surrounded by GCAS (light grey sections). Black, pores or preparation damage. (**B**) Development of the Mg/Al ratio from the center to the rim of a spinel grain along the red line in Figure 5A.

### 4.3.3. Ca-Alumosilicate (GCAS/MCAS)

The matrix component of the melt experiments (e.g., Figure 3 or Figure 4A light grey sections) can be generally expressed as $X_2YZ_2O_7$, where X can be $Na^+$ and $Ca^{2+}$; Y can be $Al^{3+}$, $Mg^{2+}$ and $Fe^{2+}$; and Z can be $Al^{3+}$ and $Si^{4+}$. The coordination of X is 8, and Y and Z are tetrahedral [23]. This Ca-alumosilicate compound generally is known as melilite. The investigated material comprises two types of Ca-alumosilicate:

- GCAS: High Al, low Si, very low Mg and virtually no Na
- MCAS: Low Al, high Si, ~3 wt.% Mg and 0.7–2.3 wt.% Na/Li is plausible

These two types are difficult to distinguish in the BSE(Z)-micrograph (Figure 4A) because of the almost same light grey shade (very similar mean atomic number). Because Na is not part of the initial materials (impurity), the concentration of the MCAS compound can be considered rather low and represents the eutectic residual melt. Nevertheless, this compound is interesting to assess a potential Li incorporation into the matrix of Ca-alumosilicate. In theory, $Li^+$ can be present in 4 or 8 coordination, whereas the ionic radius is very similar to Mg (4-coordination) or Na (8-coordination) (e.g., the ionic radii are published by Shannon (1976) [24]). The MCAS possesses a lower total sum of the measured concentrations and excess Si when calculating the chemical formula of the MCAS using the general melilite based on seven oxygen atoms.

Table 6a depicts how a calculation of virtual Ca-alumosilicate (CAS) can be conducted using five virtual compounds, namely $Li_2Si_3O_7$, $Na_2Si_3O_7$, $Ca_2Al_2SiO_7$ $Ca_2MgSi_2O_7$ and $Ca_3Si_2O_7$, assuming (limited) solid solution between those compounds. The Li value resulting from the multiplications was used to calculate a chemical formula of the analyzed MCAS. According to this calculation, the $Li_2Si_3O_7$ makes up about 1 wt.% of the total composition (see Table 6b).

**Table 6.** (**a**) Multiplication Factors (Mult.) for Calculation of an Optimized GCAS and MCAS. (**b**) Average Composition of the GCAS and MCAS Ca-Alumosilicate Solid Solution in Single Point Analysis, Compared with the Optimized Compounds Calculated with the Factors of Table 6a. Opt., Calculated Ideal Composition; Meas., Measured Average; (Calc.), Calculated Values (Li, O); %StdDev, Percentage Standard Deviation of the Measured Points (Repeats, *n* = 7 (GCAS), *n* = 6 (MCAS)).

| wt.% | Virtual Compounds | | | | |
|---|---|---|---|---|---|
| | $Li_2Si_3O_7$ | $Na_2Si_3O_7$ | $Ca_2AL_2SiO_7$ | $Ca_2MgSi_2O_7$ | $Ca_3Si_2O_7$ |
| Al | 0.0 | 0.0 | 19.7 | 0.0 | 0.0 |
| Mg | 0.0 | 0.0 | 0.0 | 8.9 | 0.0 |
| Ti | 0.0 | 0.0 | 0.0 | 0.0 | 0.0 |
| Mn | 0.0 | 0.0 | 0.0 | 0.0 | 0.0 |
| Fe | 0.0 | 0.0 | 0.0 | 0.0 | 0.0 |
| Ca | 0.0 | 0.0 | 29.2 | 29.4 | 41.7 |
| K | 0.0 | 0.0 | 0.0 | 0.0 | 0.0 |
| Si | 40.1 | 34.8 | 10.2 | 20.6 | 19.5 |
| Na | 0.0 | 19.0 | 0.0 | 0.0 | 0.0 |
| O (Calc.) | 53.3 | 46.2 | 40.8 | 41.1 | 38.8 |
| Li (Calc.) | 6.6 | 0.0 | 0.0 | 0.0 | 0.0 |
| | Multiplicator | | | | |
| GCAS | 0.00 | 0.00 | 0.98 | 0.03 | 0.00 |
| MCAS | 0.092 | 0.084 | 0.40 | 0.33 | 0.093 |
| Sum | 100 | 100 | 100 | 100 | 100 |

(**a**)

| wt.% | GCAS (Opt.) | GCAS (meas.) | GCAS %StDev. | MCAS (Opt.) | MCAS Meas. | MCAS %StDev. |
|---|---|---|---|---|---|---|
| Al | 19.2 | 19.2 | 2.8 | 7.87 | 7.87 | 10.7 |
| Mg | 0.3 | 0.3 | 46.0 | 2.94 | 2.94 | 5.2 |
| Ti | 0.0 | 0.0 | n. a. | 0.00 | 0.01 | n. a. |
| Mn | 0.0 | 0.0 | n. a. | 0.00 | 0.02 | n. a. |
| Fe | 0.0 | 0.1 | n. a. | 0.00 | 0.07 | n. a. |
| Ca | 29.4 | 29.4 | 0.4 | 25.25 | 25.25 | 4.0 |
| K | 0.0 | 0.0 | n. a. | 0.00 | 0.02 | n. a. |
| Si | 10.6 | 10.2 | 4.4 | 19.29 | 19.29 | 3.5 |
| Na | 0.0 | 0.0 | n. a. | 1.59 | 1.59 | 23.5 |
| O (Calc.) | 41.1 | 40.6 | n. a. | 42.25 | 42.29 | n. a. |
| Li (Calc.) | 0.0 | 0.0 | n. a. | 0.61 | 0.61 | n. a. |
| Sum | 100.6 | 99.7 | | 99.8 | 100.0 | |

(**b**)

The calculated formulas of the GCAS and the MCAS are:

$$\text{GCAS:} \quad Ca_{2.02}(Al_{1.96}Mg_{0.03})(Al_{1.96}Si)O_7$$

$$\text{MCAS:} \quad (Na_{0.18}Ca_{1.67}Li_{0.15})(Al_{0.52}Mg_{0.32}Li_{0.08})(Al_{0.17}Si_{1.82})O_7$$

*4.4. Comparison of Experimental Findings with Thermodynamically Modeled Subsystems*

Based on the respective phases of interest, the relevant subsystems are $MgO$-$Al_2O_3$ and $Li_2O$-$Al_2O_3$-$SiO_2$. The modeled phase diagrams are presented in Figures 6 and 7, respectively. In Figure 6, a comparison between the modeled phase equilibria and the experimental data is made. In Figure 6, the composition of the initial melt and the composition of different spinel grains from two line scans and several single spot measurements, analyzed experimentally at room temperature (RT), are presented in an overlay with the thermodynamic phase equilibrium data for the subsystem $MgO$-$Al_2O_3$. The composition of the initial melt is the starting point of the spinel crystallization. It can

be seen that all measured spinel grains show a significant higher Mg concentration compared to the initial Mg concentration in the melt.

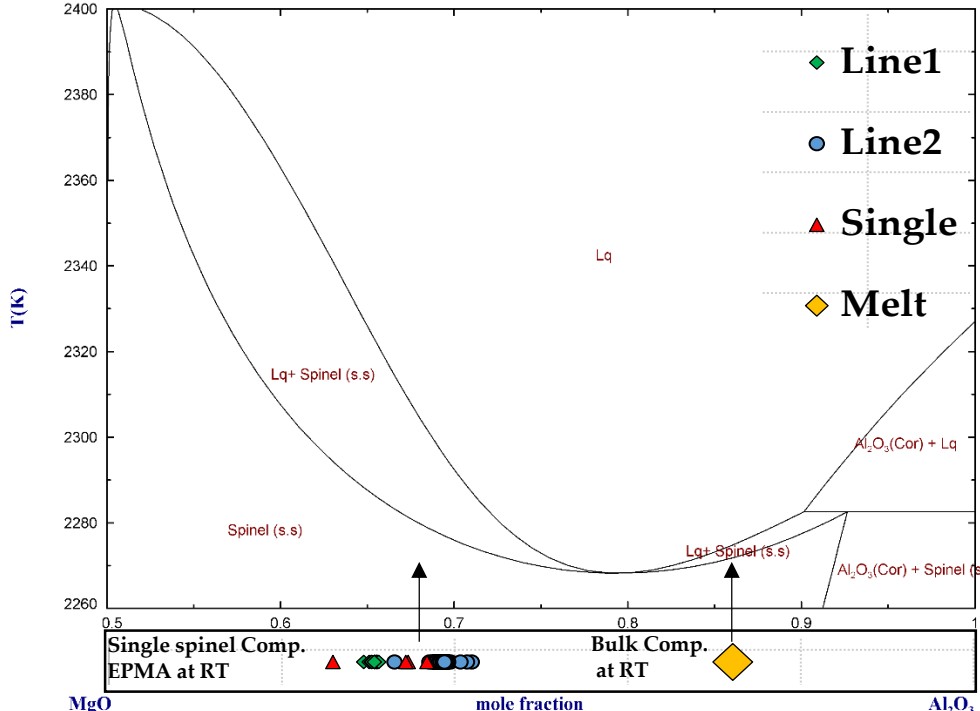

**Figure 6.** The calculated MgO-Al$_2$O$_3$ phase diagram at 1 atm total based on [22]. Lq, Liquid; spinel (s.s), spinel solid solution; Al$_2$O$_3$(Cor), corundum. In this diagram, the composition of the initial melt and the composition of different spinel grains from two line scans and several single spot measurements, analyzed at room temperature (RT), are presented. The composition of the initial melt is the starting point of the spinel crystallization.

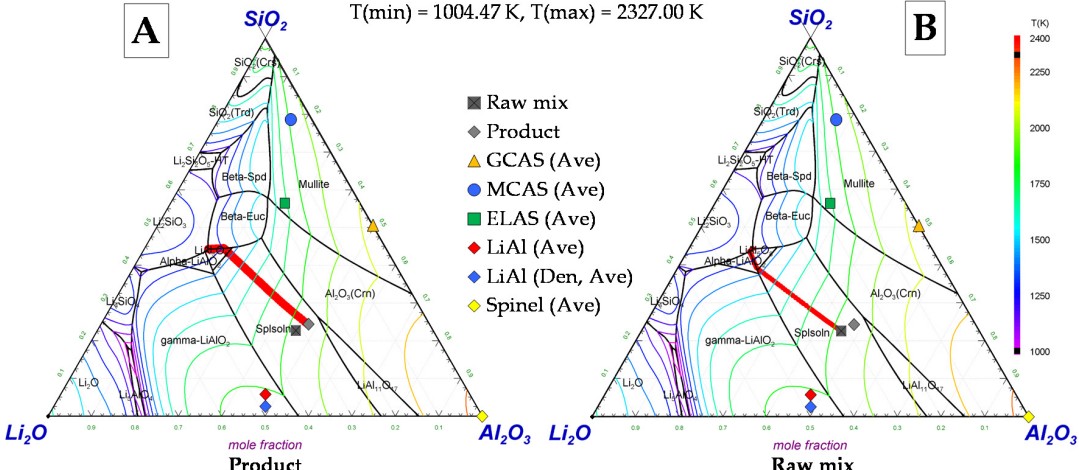

**Figure 7.** Calculated Li$_2$O-Al$_2$O$_3$-SiO$_2$ liquidus projection at 1 atm total pressure based on [22] is shown. Red line, equilibrium solidification paths starting at the initial point of the "product" (**A**) and the "raw mix" (**B**). The initial points represent the respective component concentrations in the liquid phase. Isothermal lines are drawn in Kelvin at every 100 K. In this diagram, the average compositions of the single compounds, analyzed with EPMA at room temperature (see Tables 4, 5 and 6b), and the bulk chemistry of the "raw mix" and the "product" are presented.

Figure 7 shows the thermodynamic calculated $Li_2O$-$Al_2O_3$-$SiO_2$ subsystem. The equilibrium solidification paths for the "raw mix" (Figure 7B) and the "product" (Figure 7A) composition are calculated and presented in the respective ternary phase diagram. Additionally, the average compositions of the single compounds, analyzed with EPMA at room temperature (see Tables 4, 5 and 6b), and the bulk chemistry of the "raw mix" and "product" are visualized in an overlay with the modeling results in Figure 7.

The "raw mix" and the "product" composition is in the spinel solid solution area, which is concluded by the thermodynamic modeling results based on the subsystem. Hence, the thermodynamic modeled solidification predicts spinel as the primary crystallizing phase (see Figure 8). After decreasing the temperature continuously under assumed equilibrium conditions, the solidifications of different phases are shown in Figure 8, for the "product" (Figure 8A) and "raw mix" (Figure 8B) initial concentrations, respectively. The thermodynamic prediction of the subsystem solidification shows that spinel as primary crystal is formed in solid solution with high temperature $LiAl_5O_8$ for both initial compositions. With progressing solidification, low temperature $LiAl_5O_8$ is also crystallizing out of solution. This finding holds true for both initial compositions. With increasing solidification progress, $LiAlO_2$ and $Li_2SiO_3$ are formed with a very low amount of eucryptite, for the "raw mix" configuration, while, for the "product" composition (Figure 8A), eucryptite is formed in a higher amount without any $LiAlO_2$.

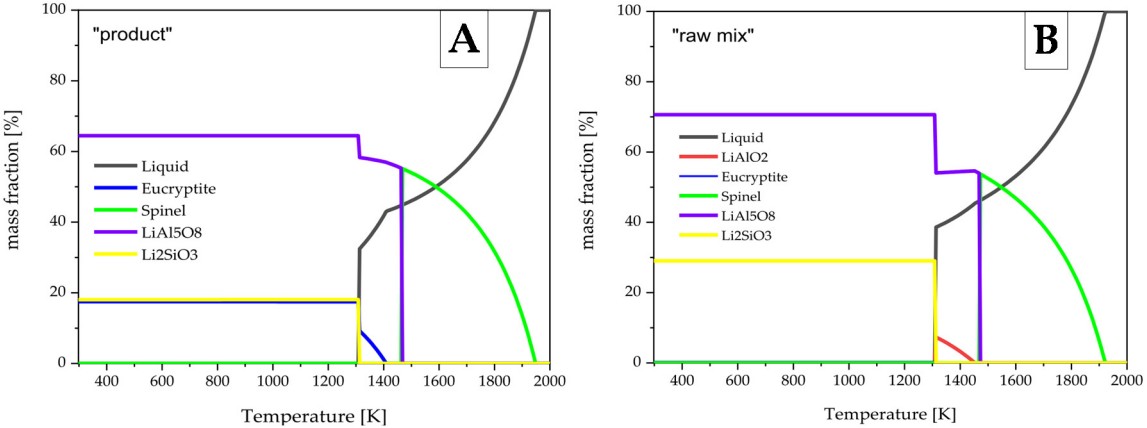

**Figure 8.** Calculated equilibrium solidification curves of the "product" (**A**) and the "raw mix" (**B**) for $Li_2O$-$Al_2O_3$-$SiO_2$ system. Calculated point interval is 5 K.

## 5. Discussion

The experimental investigation of solidified melt in connection with thermodynamic modeling of chemical reactions and solidification is an important tool to investigate how a slag system behaves and how it can be engineered. These investigations and the obtained results can serve as starting point for understanding efficient design of experiments to generate the desired phases. With a combination of thermodynamic calculation and mineralogical investigation, the probability that the artificial slag contains the desired phases can be maximized. Therefore, one purpose of the experiments carried out in this project was a first survey of the mineral compounds and the morphology of a solidified melt with the basic components $Li_2O$, $MgO$, $Al_2O_3$, $SiO_2$ and $CaO$ with a melt composition in the primary crystallization field of spinel in the subsystem $Li_2O$, $Al_2O_3$ and $SiO_2$. Another purpose was to investigate the influence of the Mg content on the ratio of the mineral compounds. The results of these experiments are also intended to serve as a basis for further thermodynamic modeling. Additionally, the applicability of the combination of PXRD and EPMA to this research topic was assessed. This includes the calculation of the lithium containing mineral compounds on basis of the EPMA result without access to measured lithium concentrations. In the following, the different identified phases are discussed:

## 5.1. Spinel-Like Oxides

The experiments show idiomorphic phenocrysts of spinel as the first crystallizing compound with decreasing temperature. The spinel crystals are surrounded by massive hypidiomorphic crystallites of LiAl and melilite-like alumosilicate (GCAS/MCAS). Additionally, LiAl forms dendritic elongated structures of hypidiomorphic crystallites. The changes in the chemistry of a single spinel crystal (Figure 5B) can help to explain a part of the crystallization curve of the melt. This is also used to validate thermodynamic model predictions of the three-component subsystem. Starting in the primary crystallization field of spinel, an Al-rich $Mg_{1-(3/2y)}Al_{2+y}O_4$ starts to form. These crystals are in equilibrium with a corresponding melt (Figure 5B (blue area) and Figure 6). The EPMA reveals that, compared with the Mg/Al ratio of the melted material, all measured spinel grains are enriched in Mg. This observation shows the complex spinel behavior, which cannot be explained with the simple binary phase diagram $MgO-Al_2O_3$. Nevertheless, the measured Mg/Al ratio increases (Figure 5B, yellow area). This can be explained with the composition of the melt reaching the phase boundary between spinel and LiAl. Through scavenging of Al from the melt during formation of LiAl, the Mg concentration in the melt increases and therefore the spinel–melt equilibrium changes. At a later stage of the crystallization process, the Al concentration in the crystal increases again, an indication that now a solid solution between $Mg_{1-(3/2y)}Al_{2+y}O_4$ and Mg-free $LiAl_5O_8$ forms (Figure 5B, green area). At the end of the crystallization, the Mg concentration rises again (Figure 5B, red area). This is an indication that the crystallization leaves the crystallization path between $LiAl_5O_8$ and spinel in direction of the crystallization path between eucryptite (or ELAS) and spinel. Therefore, the crystallization of the spinel would no longer include the aluminum-rich $LiAl_5O_8$ and the relative Mg concentration of the crystallizing spinel would increase, although a part of the Mg is incorporated into the ELAS. The crystallization path concluded by experimental observations of the developing spinel composition is on principal in good correlation with the thermodynamically predicted solidification phases in the early stages (Figures 7 and 8). However, for lower temperatures, the solidification predictions deviate from the experimental findings, which is due to non-equilibrium cooling conditions and hence phase generation. The modeled results show that small deviations in the initial concentration in the spinel solid solution field can result in strong deviations regarding appearing solid phases and solidification path behavior.

The PXRD patterns of the investigated melts with increasing Mg concentration show a displacement of the spinel main peak. The angular position of the main peak of these spinel variations is between the simple $MgO \times Al_2O_3$ compound and a pattern of an Al-rich spinel with the formula $Mg_{0.39}Al_{2.41}O_4$ and weakly correlates with the Mg concentration as:

$$Mg_y = 20.198 \times d_{311} - 48.247 \tag{1}$$

## 5.2. LiAl and ELAS

The formation of dendrites is an indication for rapid crystallization of LiAl in a small temperature interval from a supercooled melt and/or (macro)segregation (for macro segregation, see, e.g., Ahmadein et al. [25]). Due to the rather long cooling cycle (two days, Figure 1), undercooling may be improbable but cannot completely be excluded. Nevertheless, it is plausible that a segregation of an Al-Li-rich melt occurs from which the first generation of LiAl crystals forms. The Si concentrations of the dendritic LiAl is lower than in the massive LiAl, indicating a different origin, thus a different melt as well (compare LiAl in Tables 4 and 5). A few parts of the massive LiAl contain small lath-shaped grains of ELAS. This compound can be derived from eucryptite and can contain up to 18 wt.% $Mg_{0.5}AlO_2$. These grains are an indication of segregation of Si- and Mg-rich phases (melt) from the Al-rich LiAl-melt, as described above. Interestingly, the representing point of this calculated compound is not located in the primary crystallization field of pure $LiAlSiO_4$. This is due to the lower calculated Li content because of the Li-free $Mg_{0.5}AlO_2$ compound.

### 5.3. Ca-Alumosilicate

The matrix of the material consists of slightly hypidiomorphic grains of Ca-alumosilicate. The morphology of the crystals indicates that the formation starts together or slightly after the beginning of the crystallization of LiAl, which itself often shows hypidiomorphic growth. The chemical composition of these Ca-alumosilicates starts with nearly ideal gehlenite (GCAS) with minute amounts of impurities such as Mg. The other type of Ca-alumosilicate (MCAS) incorporates higher amounts of impurities such as Na and Mg (Table 6b). Because of the presence of an alkaline element such as Na, the latter compound seems to represent the end of the crystallization, i.e., the residual eutectic melt. Interestingly, this compound delivers a total of distinctively less than 100 wt.% (element concentrations calculated as simple oxide compounds) and possesses an excess of Si after calculation of the melilite formula. This is an indication that another silicious component is present in the crystal structure. Because the sample contains no free $SiO_2$ (like quartz) and the analysis shows no additional element for calculation of a silicious component, incorporation of Li into the crystal or glassy structure as shown in Table 6b is plausible. After incorporation of a virtual compound $Li_2Si_3O_7$, a formula can be calculated indicating a consistent crystal-like chemistry or a stoichiometric glass.

The mineralogical characterization of a melt as presented above provides a basis for refining the thermodynamic model, showing the real assemblage of components and the real chemical composition of the compounds/phases. An example would be ELAS. The eucryptite compound, used for the thermodynamic calculation, is ideal $LiAlSiO_4$. EPMA reveals that the real eucryptite-like alumosilicate (ELAS) can be expressed with $(Li_{0.96}Mg_{0.24})(Al)(Al_{0.45}Si_{1.55})O_6$. This compound contains Mg, which has to be taken into account when using this compound to predict a crystallization curve. The same is valid for the lithium aluminate compound (LiAl, $Li_{1-x}(Al_{1-x}Si_x)O_2$) that contains Si. Another important property is the inherent potential kinetic inhibition of the phase reactions in the system of interest. The morphology of the slag including structure and habitus corresponds to the crystallizing reactions during the cooling process. Additionally, the total chemistry and the spatial resolved development of element ratios can be used to explain the solidification process.

### 6. Conclusions and Outlook

In this work, an experimental investigation of a $Li_2O$-MgO-$Al_2O_3$-$SiO_2$-CaO system was carried out in combination with thermodynamic modeling of relevant subsystems. Based on bulk chemistry analysis, PXRD and EPMA, the crystallization paths of various phases were reconstructed and explained. It was shown that spinel is always the primary crystallizate. Furthermore, depending on minute variations in the chemistry of the melt, the result of the thermodynamically predicted further phase development can be substantially different. In this case, comparing the solidification of the raw material and the product the unpredictable loss of Li during the melt experiment seems to offer the possibility of a complete suppression of the $LiAlO_2$ formation in favor of $Mg_{1-(3/2y)}Al_{2+y}O_4$/$LiAl_5O_8$ solid solution, although this was not observed in the experiments. The eucryptite and Li-silicate compound are the ends of the solidification in both scenarios. Nevertheless, a knowledge and/or control of all reaction parameters such as partial pressures of all elements (particularly, Li here) and compounds, grain size distribution, morphology and chemistry of the raw material is crucial to develop an efficient and reproducible slag modification process. The solidification route of the system could be qualitatively predicted by thermodynamic modeling of the $Li_2O$-$Al_2O_3$-$SiO_2$ subsystem with the result that minute variations of the initial chemistry can lead to different solidification paths.

Additionally, a relative Mg enrichment of spinel grains could be observed experimentally. Furthermore, the development of the composition in single spinel grains during spinel grains give an indication of the existing solid solution $Mg_{1-(3/2y)}Al_{2+y}O_4$/$LiAl_5O_8$, which was only predicted and not verified in the past.

The results presented in this article show that Li cannot be incorporated into a single early crystallizing compound in an easy way. The investigations show that Li is present in LiAl, ELAS and with higher uncertainty in spinel (as solid solution $Mg_{1-(3/2y)}Al_{2+y}O_4$/$LiAl_5O_8$) and MCAS.

To modify this complex multi-component system (oxides of Li, Mg, Al, Si and Ca) to gain desired mineral compounds requires, besides experimental work (melt experiments, component printing and combinatorial thin film deposition), new thermodynamic modeling strategies even for higher component systems, especially with a good quantitative predictability for the phase fractions.

Furthermore, future research work will concentrate on the development of phase separation processes, predominantly by flotation for the main identified Li-bearing phases described in this paper (basic research on the way) and on the extension of component mixtures in the slag building process, advancing step by step into slag systems expected in melting processes of actual and future battery systems.

**Author Contributions:** T.S. conceived the paper. T.S., H.L. and M.F. conducted the literature review. All experiments were designed and performed by H.Q. and D.G. The chemical bulk analysis was executed by the analysis laboratory of the institute. The phase analysis (PXRD and EPMA) and the mineralogical investigation were conducted by T.S. Thermodynamic modeling was conducted by H.L. and M.F. Interpretation and discussion were conducted by all authors. All authors have read and agreed to the published version of the manuscript.

**Funding:** This research was funded by the Clausthal University of Technology in the course of a joint research project "Engineering and processing of Artificial Minerals for an advanced circular economy approach for finely dispersed critical elements" (EnAM).

**Acknowledgments:** We acknowledge support by Open Access Publishing Fund of Clausthal University of Technology.

**Conflicts of Interest:** The authors declare no conflict of interest. The funders had no role in the design of the study, in the collection, analyses, or interpretation of data, in the writing of the manuscript, or in the decision to publish the results.

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
