# Peer review of "Li-Distribution in Compounds of the Li2O-MgO-Al2O3-SiO2-CaO System—A First Survey"

_metals, doi:10.3390/met10121633_

Round 1

Reviewer 1 Report

The article reports research on the impact of spinel on the formation of separate LiAlO2 crystals by using three slags from the LiO2-MgO- Al2O3-CaO-SiO2 system.

The manuscript is interesting and contributes with original research but some corrections are needed. I suggest some other minor changes:

Along all the document:The mineral names must be written in lowercase letters.

Title. This should be “… Li2O-MgO-Al2O3-SiO2-CaO system…” or “… Li2O-MgO and Al2O3-SiO2-CaO systems…”

Line 27. Please, write the oxides of the system always in the same order.

Line 40. Pyrometallurgical is only one word.

Line 91 and in other places you begin a sentence with And. Delete the dot or delete the preposition “And”.

Line 92 In this systems two mineral phases occur: alushite and swinfordite (Brigatti et al. 2011, European Mineralogical Union Notes in Mineralogy, 11, 1-71).

Line 103. LiAlO2 and LiAl5O8 CANNOT BE polymorphous. This is a wrong interpretation of the polymorphism concept. Polimorph phases not only have the same components but the same formula. I think that you wanted to say that both have polimorphs.  

Line 104. Replace @ by the convenient Greek letter. LiAlO2 has four polimorphs: a tetragonal γ-phase, a rombohedral α-phase, an orthorhombic β-phase, and two phases of high T (li et al. 2004, J Solid State Chem. 177(6), 1939-1943.  Cubic phase?.

Line 118 and next. Replace @ by the convenient Greek letter.

Line 142. Delete the last point.

Lines 146-150. It should be better that you did first a differential thermal analysis (DTA) of the mixtures and later to program the heating paths. The melting temperature of a compound can vary slightly.

This also will be useful for all the interpretation of this research.

Line 166. The first time that you use an acronym this has also be written in full electron microprobe analysis (EPMA).

Line 171-176. Indicate which elements did you analyze with EMPA.

Line 183. “Li can be calculated as balance to identify the phases”. What do you want to say with balance?, explain this. If this is to close at 100 this is a wrong assumption. I understand in this way because later, in tables all sum 100. Note that the Li-free phases gehlenite and melilite do not close at 100 (Table 6). You never can interpret that the analysis will be perfect and close at 100.00.

The balance that you can use to calculate Li is according to the structural formula of the analysed compound.

Line 264. Gehlenite is an end member of the melilite group. Then, you have to denominate the compounds with other mane , or just to explain that the general name of melilite will be used for the phases with other cations in addition to Ca.

Line 335. Why you consider akermanite?. This was not mentioned before in your manuscript.As you sue know, akermanite and gehlenite constitute a complete solid solution and it is very difficult to distinguish by XPRD then, the name mililite group is usually used.

Line 345. Average composition of the Gehlenite (GCAS) and the Melilite (MCAS) in single point analysis” remember to use this names properly.

Author Response

Dear Reviewer 1,

This answer contains the description of the requested revisions of our manuscript:

 Li-Distribution in compounds of the Li2O-MgO- Al2O3-SiO2-CaO system – a first survey

By Thomas Schirmer (corresponding author) and Hao Qiu, Haojie Li, Daniel Goldmann and Michael Fischlschweiger

First of all we would like to thank you for elaborately reviewing our paper.

This was very helpful and we learned a lot in terms to improve our future work. We hope the revision is according to your suggestions and our manuscript can be moved to the next processing step.

Thank you for your time and consideration.

Sincerely,

Thomas Schirmer

This is the point-by-point response check-list of all changes in the document according to your comments:

Along all the document:The mineral names must be written in lowercase letters.

Changed, checked all lowercase issues

Title. This should be “… Li2O-MgO-Al2O3-SiO2-CaO system…” or “… Li2O-MgO and Al2O3-SiO2-CaO systems…”

Changed to Li-Distribution in compounds of the Li2O-MgO-Al2O3-SiO2-CaO system – a first survey

Line 27. Please, write the oxides of the system always in the same order.

Changed

Line 40. Pyrometallurgical is only one word.

Changed

Line 91 and in other places you begin a sentence with And. Delete the dot or delete the preposition “And”.

This is a screenshot of the original text:

Not clear what to change – word (srtg+h) didn’t find And.

Line 92 In this systems two mineral phases occur: alushite and swinfordite (Brigatti et al. 2011, European Mineralogical Union Notes in Mineralogy, 11, 1-71).

Thank you for your comment – this minerals are really not easy to find

Alushite: Na0.5(Al,Mg)6(Si,Al)8O18(OH)12·5H2O (https://www.mineralienatlas.de/lexikon/index.php/MineralData?lang=de&mineral=Alushit) and swinfordite (Handbook of Clay Science, Lagaly and Bergaya, Elsevier ASIN : B00D6LVZPY), (Li,Al,Mg)2.5[(OH,F)2/(Si3.8Al0.2)O10]·(Li,Ca,…)0.3·(H2O)4 (e. g. Strunz, H. Nickel, E.H., Strunz – Mineralogical Tables, Schweizerbart) seem to be clay minerals(Montmorillonit/Vermiculit-group) – in this high temperature and water-free environment they are instable

Line 103. LiAlO2 and LiAl5O8 CANNOT BE polymorphous. This is a wrong interpretation of the polymorphism concept. Polimorph phases not only have the same components but the same formula. I think that you wanted to say that both have polimorphs.

This is correct - changed

Line 104. Replace @ by the convenient Greek letter. LiAlO2 has four polimorphs: a tetragonal γ-phase, a rombohedral α-phase, an orthorhombic β-phase, and two phases of high T (li et al. 2004, J Solid State Chem. 177(6), 1939-1943.  Cubic phase?

Did miss this – changed – Reference list actualized (all from 9 +1)

The cubic I4132 phase is documented here: Konar, B.; Van Ende, M.-A.; Jung, I-H. Critical Evaluation and Thermodynamic Optimization of the Li2O-Al2O3 and Li2O-MgO-Al2O3 Systems. Metallurgical and Materials Transactions B 2018, 49, 5: 2917–44. https://doi.org/10.1007/s11663-018-1349-x.

Line 118 and next. Replace @ by the convenient Greek letter.

General formatting problem – greek symbols actualized throughout the whole document

Line 142. Delete the last point.

changed

Lines 146-150. It should be better that you did first a differential thermal analysis (DTA) of the mixtures and later to program the heating paths. The melting temperature of a compound can vary slightly.

Good point – I have to apologize that in this stadium of the article review we cannot switch in DTA, but we will consider this in future

This also will be useful for all the interpretation of this research.

Good point (s. a.)

Line 166. The first time that you use an acronym this has also be written in full electron microprobe analysis (EPMA).

Changed (s. a.)

Line 171-176. Indicate which elements did you analyze with EMPA.

Changed

Line 183. “Li can be calculated as balance to identify the phases”. What do you want to say with balance?, explain this. If this is to close at 100 this is a wrong assumption. I understand in this way because later, in tables all sum 100. Note that the Li-free phases gehlenite and melilite do not close at 100 (Table 6). You never can interpret that the analysis will be perfect and close at 100.00.

Changed to … Li can be calculated using virtual compounds as depicted in Table 4.

The balance that you can use to calculate Li is according to the structural formula of the analysed compound.

Changed (s. a.)

Line 264. Gehlenite is an end member of the melilite group. Then, you have to denominate the compounds with other mane , or just to explain that the general name of melilite will be used for the phases with other cations in addition to Ca.

Changed: added to the list:  The compound (GCAS) is an end member of the melilite-like calcium-alumosilicate (MCAS), which is used for this phase with higher amounts of ions in addition to Ca

Line 335. Why you consider akermanite?. This was not mentioned before in your manuscript. As you sue know, akermanite and gehlenite constitute a complete solid solution and it is very difficult to distinguish by XPRD then, the name mililite group is usually used.

Changed: removed the mineral names from the virtual compounds used to calculate the Ca-alumosilicate solid solution.

Line 345. Average composition of the Gehlenite (GCAS) and the Melilite (MCAS) in single point analysis” remember to use this names properly.

Changed: Average composition of the (GCAS) and the (MCAS) Ca-alumosilicate solid solution

Reviewer 2 Report

The article is important because it brings new fundamental information on the recovery of important elements from lithium batteries by pyrometallurgical techniques. Of course the research must be continued / completed and compared with the hydrometallurgical alternative.

Author Response

Dear Reviewer 2,

This letter contains the description of the requested revisions of our manuscript:

 Li-Distribution in compounds of the Li2O-MgO- Al2O3-SiO2-CaO system – a first survey

By Thomas Schirmer (corresponding author) and Hao Qiu, Haojie Li, Daniel Goldmann and Michael Fischlschweiger

First of all we would like to thank you for elaborately reviewing our paper.

Thank you for your time and consideration.

Sincerely,

Thomas Schirmer

Thank you very much vor the very good rating of our article.

Nothing to change

Reviewer 3 Report

Dear Authors,
I have read Your manuscript with great attention and interest. In my opinion, the paper fills the gap existing in described subject and gives very important information for scientists and engineers. The submission falls within the scope of the Metals journal and is sufficiently original and comprehensive and I recommend it to publish after introduction of the following changes:

Line 149,  500°C  value and unit should be separated

Line 159, note as above

It seems to me, that the quality and resolution of pictures 7 and 8 should be higher.

Additionally, in the footer of the manuscript the name of the journal "Minerals" is incorrect.

Generally, research and results are well presented and commented on. I suggested some small editorial changes in the paper, so I recommend to publish the manuscript after MINOR REVISIONS.

Regards,

Author Response

Dear Reviewer 3,

This letter contains the description of the requested revisions of our manuscript:

 Li-Distribution in compounds of the Li2O-MgO- Al2O3-SiO2-CaO system – a first survey

By Thomas Schirmer (corresponding author) and Hao Qiu, Haojie Li, Daniel Goldmann and Michael Fischlschweiger

First of all we would like to thank you for elaborately reviewing our paper.

This was very helpful and we learned a lot in terms to improve our future work. We hope the revision is according to your suggestions and our manuscript can be moved to the next processing step.

Thank you for your time and consideration.

Sincerely,

Thomas Schirmer

Point-by-point response to all reviewer requests:

Line 149,  500°C  value and unit should be separated

Changed

Line 159, note as above

Changed

It seems to me, that the quality and resolution of pictures 7 and 8 should be higher.

Changed for better res. Max Quality limited to thermodynamical modeling software output

Additionally, in the footer of the manuscript the name of the journal "Minerals" is incorrect.

Changed

Round 2

Reviewer 1 Report

The changes that I suggested were introduced and now i find the manuscript ready to be accepted for publication.